# A Systematic Study of the Antibacterial Activity of Basidiomycota Crude Extracts

**DOI:** 10.3390/antibiotics10111424

**Published:** 2021-11-21

**Authors:** Marco Clericuzio, Mattia Bivona, Elisa Gamalero, Elisa Bona, Giorgia Novello, Nadia Massa, Francesco Dovana, Emilio Marengo, Elisa Robotti

**Affiliations:** 1Dipartimento di Scienze e Innovazione Tecnologica, Università del Piemonte Orientale, Viale T. Michel 11, 15121 Alessandria, AL, Italy; marco.clericuzio@uniupo.it (M.C.); mattia.bivona@uniupo.it (M.B.); elisa.gamalero@uniupo.it (E.G.); giorgia.novello@uniupo.it (G.N.); nadia.massa@uniupo.it (N.M.); emilio.marengo@uniupo.it (E.M.); elisa.robotti@uniupo.it (E.R.); 2Dipartimento di Scienze e Innovazione Tecnologica, Università del Piemonte Orientale, Piazza San Eusebio 5, 13100 Vercelli, VC, Italy; 3Via Quargnento 17, 15029 Solero, AL, Italy; francescodovana@gmail.com

**Keywords:** *Cortinarius*, *Mycena*, *Ramaria*, antifungal activity, antibacterial activity, *Candida albicans*, *C. glabrata*, *Staphylococcus aureus*, *Pseudomonas aeruginosa*, *Klebsiella pneumoniae*

## Abstract

The excessive consumption of antibiotics in clinical, veterinary and agricultural fields has resulted in tremendous flow of antibiotics into the environment. This has led to enormous selective pressures driving the evolution of antimicrobial resistance genes in pathogenic and commensal bacteria. In this context, the World Health Organization (WHO) has promoted research aiming to develop medical features using natural products that are often competitive with synthetic drugs in clinical performance. Fungi are considered an important source of bioactive molecules, often effective against other fungi and/or bacteria, and thus are potential candidates in the search of new antibiotics. Fruiting bodies of sixteen different fungal species of Basidiomycota were collected in the Italian Alps. The identification of fungal species was performed through Internal Transcribed Spacer (ITS) sequencing. Most species belong to genera *Cortinarius*, *Mycena* and *Ramaria*, whose metabolite contents has been scarcely investigated so far. The crude extracts obtained from the above mushrooms were tested for their inhibition activity against five human pathogens: *Candida albicans* ATCC 14053, *C. glabrata* ATCC 15126, *Staphylococcus aureus* NCTC 6571, *Pseudomonas aeruginosa* ATCC 27853 and *Klebsiella pneumoniae* ATCC 13883. Twelve crude extracts showed activity against *P. aeruginosa* ATCC 27853. Highest activity was shown by some *Cortinarius* species, as *C. nanceiensis*.

## 1. Introduction

Once antibiotic pressure is established by the overuse and misuse of antibiotics, bacteria can react through different ways including drug inactivation/modification, binding sites/targets modifications, alteration of the cell permeability [1]. Moreover, biofilm formation covered with an exopolysaccharide layer, ensuring protection against antibiotics or immune system activity, can develop if the bacterial cells are able to release and perceive signal molecules leading to a bacterial communication process via quorum sensing [2]. All these survival strategies lead to antimicrobial resistance (AMR) that is currently one of the most serious threat to public health worldwide. AMR is expensive both considering human health and social cost. In fact, infections related to AMR bacteria imply an increased negative outcome for the patient, longer hospital stay, and higher morbidity and mortality rate. According to the last report of the Organization for Economic Co-operation and Development (OECD) released in 2018 (https://www.oecd.org/health/stemming-the-superbug-tide-9789264307599-en.htm (accessed on 9 November 2021)), antidrug resistant bacteria cause about 60,000 deaths each year, of which 33,000 are in the EU/EEA Countries, and 29,500 are in the United States. The projection model estimates that by 2050, 2.4 million people will die due to infections carried out by antibiotic resistant bacteria. Together with such a terrifying scenario, the economic costs of AMR reaching around USD 3.5 billion per year must be considered. In order to face AMR, a Global Action Plan was released in 2015 by the World Health Organization (WHO). In this context, a list of six bacterial species against which it is mandatory to discover and develop new drugs has been established [3]. The acronym ESKAPE encloses these human pathogens, typically associated with nosocomial infections and deserving most attention under a clinical point of view: *Enterococcus faecium*, *Staphylococcus aureus*, *Klebsiella pneumoniae*, *Acinetobacter baumannii*, *Pseudomonas aeruginosa*, and *Enterobacter* species [4,5]. While encompassing Gram-positive and -negative bacterial species, the microorganisms classified as ESKAPE share the capability to “escape” the biocidal effects of antibiotics. To give an idea, according to the latest report of the European Centre for Diseased Prevention and Control (ECDC) (https://www.ecdc.europa.eu/sites/default/files/documents/surveillance-antimicrobial-resistance-Europe-2019.pdf (accessed on 9 November 2021)), in 2019, more than a third of the *K. pneumoniae* isolates were resistant to at least one antimicrobial group under surveillance, and often characterized by a combined resistance to different antimicrobial groups. In several countries, the rate of carbapenem resistance percentages was above 10% in *K. pneumonia* and *P. aeruginosa*, with Italy reaching over 35%. On the contrary, although a reduction in the percentage of methicillin-resistant (i.e., MRSA) *S. aureus* strain, MRSA is always considered an important pathogen in European countries, also showing combined resistance to other antimicrobial groups.

Besides bacterial species, yeast such as *Candida* spp. cause common yeast infections, which can affect the mouth, skin, and vagina, but also become systemic resulting in more than 3.6 million U.S. healthcare visits each year, and USD 3 billion estimated direct medical costs [6]. While most *Candida* infections are caused by the species *Candida albicans*, showing very low levels of antibiotic resistance, other species including *Candida glabrata* and *Candida auris* are frequently resistant and more deadly [7]. The therapy against yeast infection is mainly based on azoles and polyenes. However, this approach is not always successful due to their fungistatic more than fungicidal effects, and to drug resistance developed by the *Candida* spp. isolates.

Fungi often produce antibacterial metabolites as a result of competition in the wild. Extracts from mycelial cultures and fruiting bodies offered new promising antibiotic compounds [8]. Fruiting bodies of species belonging to phylum Basidiomycota (basidiomes) collected in the wild, are the object of the present investigation. Various screenings of basidiomes for antibiotic activity have recently appeared in the literature [9,10,11,12,13] and several extracts have shown significant results.

In this work, we assessed the antibacterial and antifungal properties of crude extracts obtained from 16 Basidiomycota species against the reference strain of five human pathogens such as *C. albicans* ATCC 14053, *C. glabrata* ATCC 15126, *S. aureus* NCTC 6571, *P. aeruginosa* ATCC 27853 and *K. pneumoniae* ATCC 13883. In particular, we have focused our research on 16 species, mostly belonging to genera *Cortinarius*, *Mycena*, and *Ramaria*. These are genera including an high number of species, that have been, to date, poorly investigated concerning their metabolite contents. The aim of this study is to identify those fungal species whose extracts can reduce or totally inhibit the growth of important human pathogens.

## 2. Results and Discussion

### 2.1. Taxonomic Classification

The fruiting bodies employed in the present investigation were collected in three different sites as listed in Table 1. The *Ramaria* sp. alignment (Figure 1) includes 37 sequences and comprised 792 characters. The SC_1_RAM sequence grouped with three sequences of *Ramaria flavescens* (GenBank: KP967538, AJ408357, KY626142; MLB = 100); SC_2_RAM sequences clustered with sequences named *Ramaria flavescens* (GenBank: MK493041, AJ408357, KY626142; MLB = 100), and the sequences of SC_5_BOT grouped in *Parabotrytis* clade (GenBank: MH216040, MH216039; MLB = 100).

The *Cortinarius* sp. alignment (Figure 2) includes 36 sequences and comprised 607 characters. The SC_3_COR sequence grouped with two sequences of *Cortinarius russeoides* (GenBank: KJ421060 and AF389136; MLB = 100), the SC_4_COR sequence clustered with three sequences of *Cortinarius percomis* (GenBank: LT797160, AY669529, NR_130242; MLB = 100). Extraction and amplification of DNA from species n. 4 failed, so we report it under the provisional name of *Cortinarius cfr. reverendissimus.*

### 2.2. Biological Activity of Fruitbody Crude Extracts

The different crude extracts did not show any growth inhibition against the two *Candida* species (*C. albicans* and *C. glabrata*) and *K. pneumoniae* and *S. aureus* ATCC strains. However, some of the extracts demonstrated biological activity, expressed as inhibition of bacterial growth and/or pyocyanin release against *P. aeruginosa* ATCC 27853. Pyocyanin is a phenazine pigment synthesized by 90–95% *P. aeruginosa* strains [14] whose production is quorum-sensing regulated, described for the first time in 1860 during the observation of a bluish pus sample [15]. The molecule is soluble in chloroform and diffusible in water. Due to its cytotoxicity, pyocyanin is considered as a virulence factor inducing oxidative stress in prokaryotes and eukaryote cells, through the flow of electrons and the accumulation of ROS after reaction with O_2_ [14]. It has been estimated that the lethal amount of this molecule against other organisms (bacteria, fungi, yeasts, protozoa, algae) varied according to the target cell and range from very little concentrations up to 2000 µg mL^−1^ [16]. As shown in Figure 3 and Figure 4, some crude extracts (n. 1, n. 10, n. 16) induced the formation of a halo that apparently was not related to inhibition of bacterial growth, but mainly to the synthesis of pyocyanin.

These two observed biological activities were often combined (as showed in Figure 3 for example) and, therefore, they were shown as combined results (Figure 4), which is useful to select the potential phytochemical producer. The positive controls were represented by the Imipenem and Meropenem inhibition activity (A and B boxes) and indicated the reference of the potential crude extract activity, while the negative controls were dioxane and only MH medium.

Figure 4 showed the variability of the biological activity induced by the extracts from the different fungal species analyzed.

In order to assess if the observed effect was due to an inhibition of pyocianin production or to a chemical modification of this pigment we settled up an experiment on CAS agar (the universally accepted medium for siderophore production detection). 

The results obtained (Figure 5 and Figure 6) showed that the siderophore synthesis on CAS agar was unaffected by crude extracts. Consequently, we hypothesized that the fungi crude extracts induced a chemical modification of pyocyanin, previously described as pyocyanin bleaching, due to a redox reaction of the molecule [17]. The same pyocyanin bleaching has been reported by Reszka et al. [18] who observed a modification of the phenazine chromophore due to the pyocyanin oxidation by H_2_O_2_. Interestingly, this modified pyocyanin were less active than the non-oxidized pyocyanin [18].

*Cortinarius* is the richest species genus of class Agaricomycetes, with some 2000 estimated species worldwide. Phytochemical investigations of *Cortinarius* basidiomes has mainly concerned chromogenic compounds: these are mostly anthraquinones and related compounds, more often resulting from the octaketidic biosynthetic route [19]. Octaketidic anthraquinones and pre-anthraquinones, both monomers (as for instance, physcion) and dimers (as flavomannin), have often shown strong antibacterial properties [20]. Beattie et al. [21] have carried out a screening of 117 species of Australian *Cortinarius* in order to assess their activity against *S. aureus* and *P. aeruginosa*: the species able to produce anthraquinoids have generally yielded more significant bacterial growth inhibition. In our case, the crude extracts of *C. mussivus* (n. 1), *C. percomis* (n. 5) and *C. nanceiensis* (n. 6), were active: all three species are known to synthesize antraquinonic compounds [19]. In particular, the extract of *C. nanceiensis* was by far the most active.

Conversely, the lack of growth inhibition by the *C. variecolor* extract (n. 3) may be due to the fact that it does not synthesize anthraquinones, but rather simple phenolics (our unpublished results), which do not present antibiotic properties against the considered microorganisms. Regardless, this extract (n. 3) showed bleaching activity on siderophores, suggesting a potential activity in the virulence inhibition. Although being unable to synthesize antraquinones, *C. caesiocanescens* (n. 2), *C. reverendissimus* (n. 4), and *C. bovinus* (n. 7), were active against *P. aeruginosa* ATCC 27853 suggesting that other molecule classes may exert important antibiotic properties. Moreover, to our knowledge, these three species have never been subjected to phytochemical analysis.

*Mycena* is another genus containing a high number of species, about 700 species worldwide. Strobilurines, a class of complex aromatic-polyenic molecules derived from mixed mevalonic and shikimic acid routes, have been isolated from mycelial cultures of some *Mycena* species, but not from basidiomes. Strobilurines have shown marked antibacterial, antifungal and cytotoxic properties [22]. Pyrroloquinoline alkaloids have been recently isolated from the basidiomes of different species of the genus *Mycena* such as *M. haematopus*, *M. sanguinolenta* [23] and *M. rosea*. One of these compounds, haematopodin B, is strongly active against the soil bacterium *Azoarcus tolulyticus* [24]. In the present work, the extracts of the two *Mycena* species investigated, i.e., *M. renati* (n. 15) and *M. zephyrus* (n. 16) were active against *P. aeruginosa*. No phytochemical analysis has been previously performed on these two species.

The precise number of species belonging to the genus *Ramaria* is difficult to estimate since only molecular analysis can yield a reliable determination and DNA fingerprinting of *Ramaria* has recently started. However, there are only few works in the literature concerning Ramaria phytochemistry. Ramariolides A-D, isolated from *R. cystidiophora*, have an insaturated γ-lactone moiety (butenolide) which, in ramariolide A forms an unusual spiro-bicyclo moiety with an oxirane ring [25]. This metabolite showed antimicrobial activity against *Mycobacterium smegmatis* and *M. tuberculosis* [25].

Here, we found that the extracts of *R. parabotrytis* (n. 9), *R. pallidosaponaria* (n. 10) and *R. flavescens* (n. 11) were all active against *Pseudomonas aeruginosa* ATCC 27853. We are unaware of any phytochemical investigation concerning these three species.

The only species outside the above three genera which showed activity was *Hydnellum spongiosipes* (n. 14), a species which was never been previously characterised by phytochemical analysis.

*Cortinarius*, *Mycena* and *Ramaria* are three species-rich genera (mainly the former two) which, however, have been relatively little investigated so far. One reason is that specific assignment of species has been considerably difficult on a morphological basis; presently DNA fingerprinting has allowed the scientific community to have a much sounder identification tool. Here, we have performed ITS sequence analysis of several species, in particular those belonging to morphologically critical groups. A second reason, more difficult to overcome, is the considerable amounts of fungal material needed to the isolation and structural characterization of unknown molecules. In fact, also as concerns our work, this was an important limiting factor.

## 3. Materials and Methods

### 3.1. Sampling and Taxonomic Classification

#### 3.1.1. Fungal Material

The fruiting bodies employed in the present investigation (Table 1, Figure 7) were collected in three different sites: species 2, 4, 5, 8, 9, 13 in sample site A (Upega, CN, Italy; 44.1232 N 7.7188 E); species 1, 3, 6, 7, 10, 11, 12, 16 in sample site B (Passo della Mendola, BZ, Italy; 46.4168 N 11.2072 E); species 14 and 15 in sample site C (Piani di Praglia, GE, Italy; 44.5150 N 8.8069 E).

#### 3.1.2. DNA Fingerprinting

Genomic DNA was extracted from dry fragments of five analyzed specimens used NaOH extraction method employed by Dovana and coworkers [26]. The nrDNA ITS region was amplified with primers ITS1F [27] and ITS4 [28]. Sequencing was performed by Biofab s.r.l. Sequences were assembled and edited in Geneious v. 11.1.5 [29], submitted to GenBank: SC3_CORT/OL375429, SC1R/OL375430, SC2R /OL375431, SC5/OL375432, SC4_COR/OL375433, and then compared to those available in GenBank database with the BLASTn algorithm. On the base of BLAST results, two ITS datasets were produced: the first includes sequences belonging to the genus *Cortinarius*, *Cortinarius vesterholtii* (DQ350842) was chosen as an outgroup and the second includes sequences belonging to the genus *Ramaria*, *Gomphus clavatus* (EU118628) was chosen as an outgroup.

The sequences were aligned using MAFFT v 7.017 [30] in Geneious v. R 11.1.5 setting auto algorithm. The Maximum Likelihood analysis was performed with RAxML v. 8.2.11 [31] implementing the GTR + G model and a total of 1000 bootstrap replicates with “Rapid Bootstrapping and search for best-scoring ML tree” option.

### 3.2. Extraction of Phytochemicals from Fruiting Bodies

The solvent used for extraction was 90% acetone/10% 2-propanol (*v/v*). This solvent system allows metabolites of a wide polarity range to be extracted in a single step, with the exception of extremely hydrophilic molecules, as aminoacids and partly sugars. MeOH or EtOH were excluded at this stage, as they are known to produce artifacts, mainly when in contact with the fungus enzymatic pool [32].

In brief, the fungal material was directly minced inside the solvent (in the constant amount of 1.5 L per kg of fresh fungal material), using a blender. The suspension was kept under stirring at rt for three hours, after which time it was filtered and evaporated under reduced pressure. The solid material was extracted for a second time with a reduced amount of new solvent overnight (0.75 L kg^−1^), then filtered and evaporated as above. The two extracts were then pooled together in a single sample.

### 3.3. Antifungal and Antibacterial Assays

The agar disc diffusion method was employed to determine the antifungal and antibacterial activity of the different fungal extract, according to the methods previously published [33,34,35,36].

#### 3.3.1. Antifungal Assay

The antifungal assays were carried out with the reference strains *C. albicans* ATCC 14053 and *C. glabrata* ATCC 15126. The antifungal effects of clotrimazole (10 μg) and extracts were evaluated according to Clinical and Laboratory Standards Institute Standard M44-A. Briefly, strain suspensions (10^6^ CFU mL^−1^) were swabbed on Mueller-Hinton Agar added with 2% Glucose and 0.5 μg/mL Methylene Blue Dye (GMB). Filter paper disc (6.0 mm diameter) were placed on the agar surface and added with 10 μL of the extract in order to evaluate its antifungal activity. Clotrimazole (10 μg) discs were used as positive control. 1,4 Dioxane (Sigma-Aldrich, St. Louis, MO, USA; 10 μL) and organic linseed oil (10 μL) discs were used as negative control. Plates were incubated at 37 °C for 48 h. All experiments were performed in triplicate. The sensitivity test for the extract is considered positive if in an inhibition halo higher than that induced by clotrimazole (positive control ≥ 100%).

#### 3.3.2. Antibacterial Assay

The antibacterial assays were carried out with the reference strain *Staphylococcus aureus* NCTC 6571, *Pseudomonas aeruginosa* ATCC 27853, *Klebsiella pneumoniae* ATCC 13883. Vancomycin, Imipenem and Meropenem effects were evaluated according to EUCAST Disk Diffusion Method for Antimicrobial Susceptibility v. 7.0 (January 2021). The sensitivity to the extracts was assessed using agar disk diffusion method: strain suspensions (0.5 McFarland), obtained in physiological solution, were swabbed on Mueller-Hinton Agar (Biolife Italiana s.r.l., Milan, Italy) plates. Filter paper disc (6.0 mm diameter) were placed on the agar surface and added with 10 μL of extracts in order to evaluate its antibacterial activity. 1,4 Dioxane (Sigma-Aldrich, St. Louis, MO, USA; (10 μL) and organic linseed oil (10 μL) disks were used as negative controls, while vancomycin, meropenem and imipenem were considered as positive control. Plates were incubated at 37 °C for 24 h. All experiments were performed in triplicate. The halos were measured in mm using calipers. The extract was considered active when produced a halo equal or higher than positive control (positive control ≥ 100%).

### 3.4. Siderophore Production

Siderophore production was evaluated on Chrome Azurol S (CAS) agar according to (Schwyn and Neilands, 1987) [37]. The bacterial strains were inoculated at the center of each plate and incubated at 28 °C for seven days. A 6 mm filter paper disk was placed on the colony and added with 10 μL of extracts or 10 μL of saline solution (0.8%) in order to evaluate the possible siderophore synthesis inhibition. The ability to produce siderophore was indicated by the occurrence of a yellow-orange halo around the colony and was measured with a caliper as the ratio between the two diameters of the halo and the two diameters of the colony.

### 3.5. Statistical Analysis

Disk diffusion results were statistically analyzed by one-way ANOVA followed by Tukey HSD multiple comparisons of means using R (v. 3.5.1) [38]. Data are presented as boxplots. Differences were considered significant for *p* values < 0.05.

## 4. Conclusions

This preliminary screening of Basidiomycota fruiting bodies for antibiotic activity showed promising results, since the crude extracts of 12 out of 16 species were active against *P. aeruginosa.* In particular, *Cortinarius nanceiensis* demonstrated the highest biological activity indicated by bacterial growth suppression and modification of the pyocyanin synthesized by *P. aeruginosa* ATCC 27853. However, this is only the first step of the work; by testing the activity of the single chromatographic fractions of the crude extracts selected in this work, we will be able to greatly improve our understanding of these data. In a second paper, presently in preparation, the attention will be shifted from crude extracts to chromatographic fractions and to single compounds, where possible. We believe that such a systematic work is a much needed one, having been performed only partially in the past.

## Figures and Tables

**Figure 1 antibiotics-10-01424-f001:**
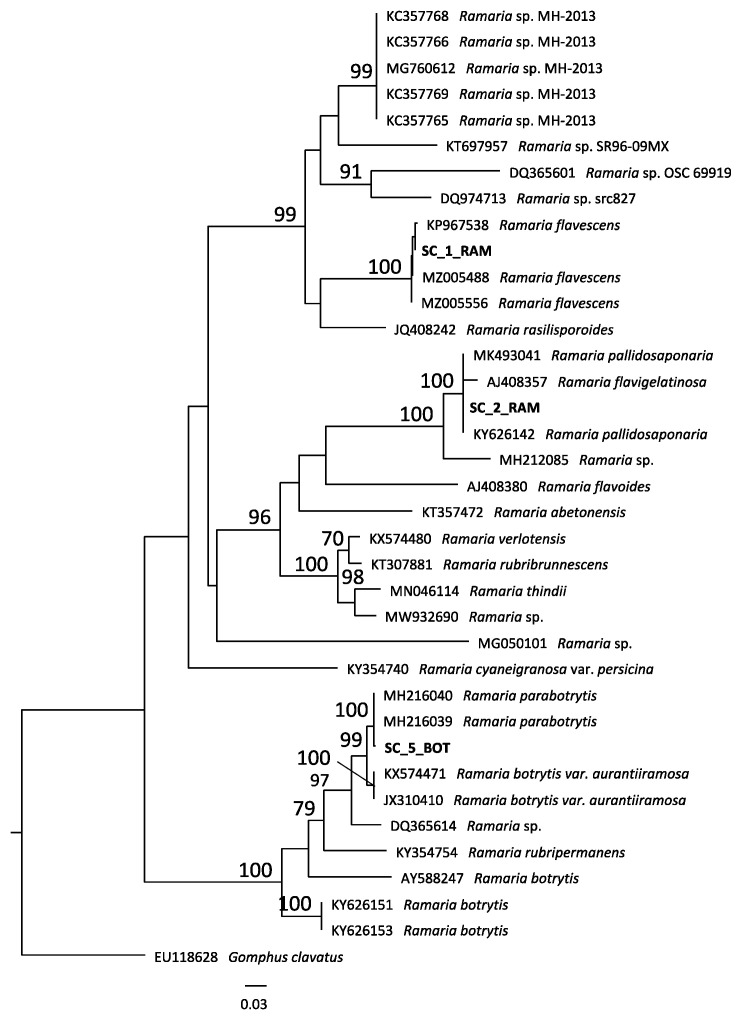
Phylogenetic tree of *Ramaria* sp.

**Figure 2 antibiotics-10-01424-f002:**
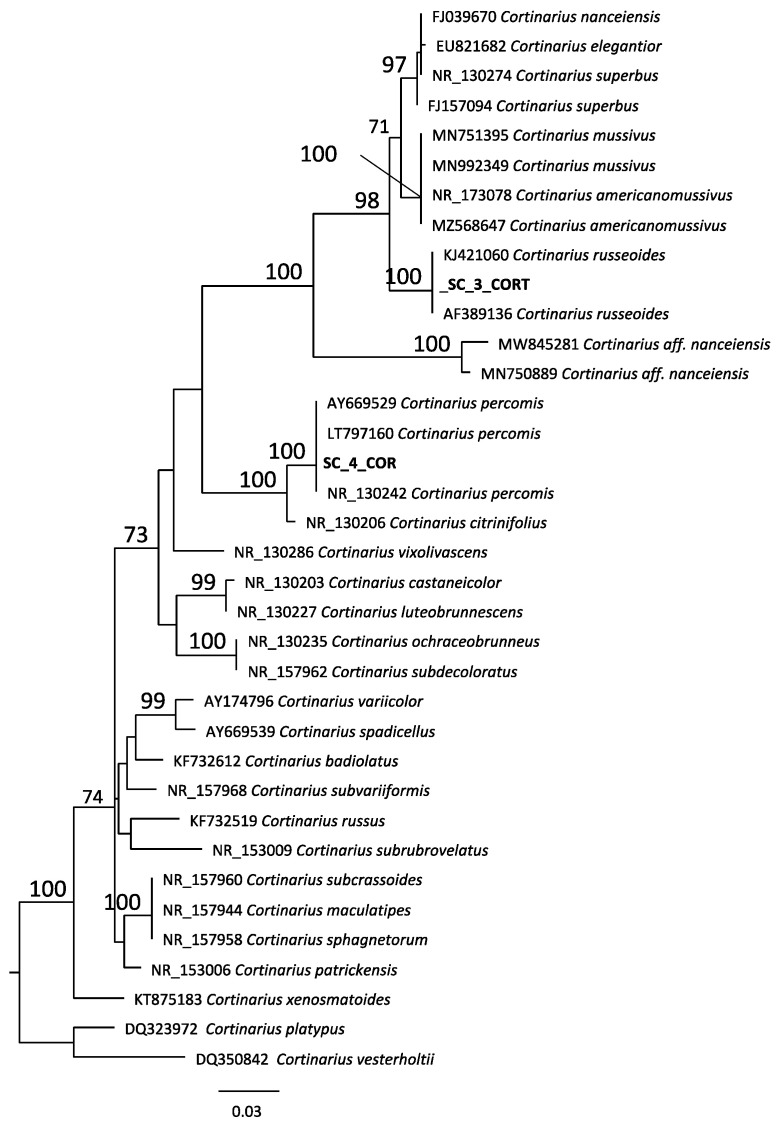
Phylogenetic tree of *Cortinarius* sp.

**Figure 3 antibiotics-10-01424-f003:**
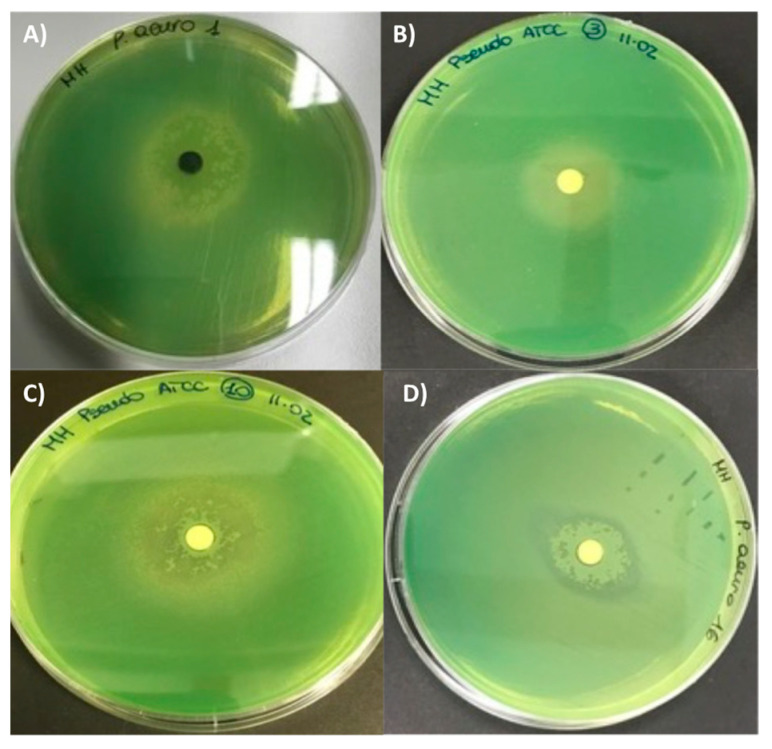
Halo (biological activity) induced by some of crude extract obtained from fruiting body of different fungi induced in *P. aeruginosa* ATCC 27853. (**A**) *Cortinarius mussivus* (n. 1); (**B**) *C. variecolor* (n. 3); (**C**) *Ramaria sanguinea* (n. 10); (**D**) *Mycena zephirus* (n. 16). Halo was measured using caliper.

**Figure 4 antibiotics-10-01424-f004:**
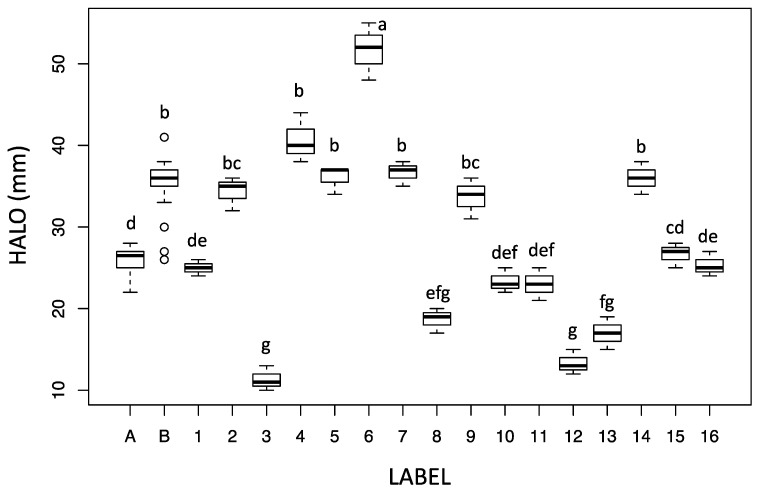
Growth and siderophore modification halo diameter (mm) (biological activity) induced by the crude extract obtained from fruiting body of different fungi induced in *P. aeruginosa* ATCC 27853. Label: A: Imipenem; B: Meropenem; 1: *Cortinarius mussivus*; 2: *C. caesiocanescens*; 3: *C. variecolor*; 4: *C. reverendissimus*; 5: *C. percomis*; 6: *C. nanceiensis*; 7: *C. bovinus*; 8: *Ramaria pallida*; 9: *R. parabotrytis*; 10: *R. pallidosaponaria*; 11: *R. flavescens*; 12: *Leucocortinarius bulbiger*; 13: *Gymnopus confluens*; 14: *Hydnellum spongiosipes*; 15: *Mycena renati*; 16: *M. zephyrus*. Different letters upon the bars indicate significant differences according to Tukey’s HSD test (*p*-value cutoff = 0.05). EUCAST Breakpoint for *P. aeruginosa*: Imipenem ≥ 50 Sensitive, <20 Resistant; Meropenem ≥ 24 Sensitive, <18 Resistant.

**Figure 5 antibiotics-10-01424-f005:**
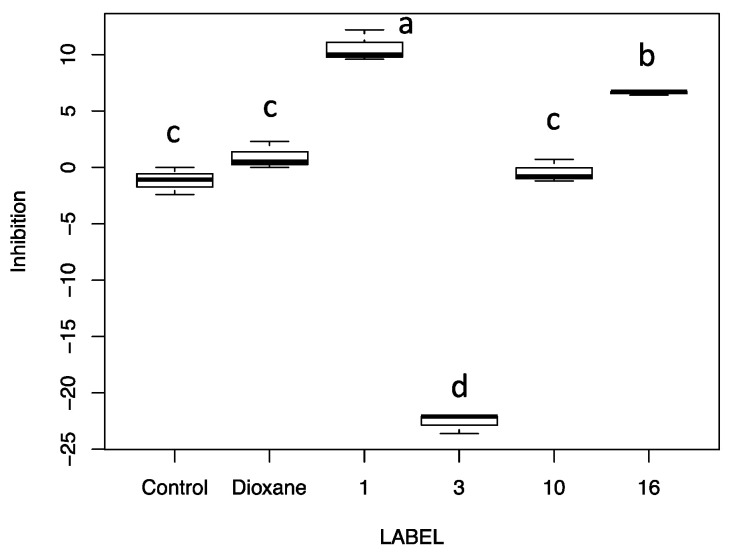
Inhibition percentage of siderophore production (biological activity) induced by the crude extract obtained from fruiting body of different fungi induced in *P. aeruginosa* ATCC 27853. Labels: Control = Negative control (only medium); Dioxane = Dioxane (negative control); 1: *Cortinarius mussivus*; 3: *C. variecolor*; 10: *Ramaria pallidosaponaria*; 16: *Mycena zephyrus*. Halo was measured using caliper. Different letters upon the bars indicate significant differences according to Tukey’s HSD test (*p*-value cutoff = 0.05).

**Figure 6 antibiotics-10-01424-f006:**
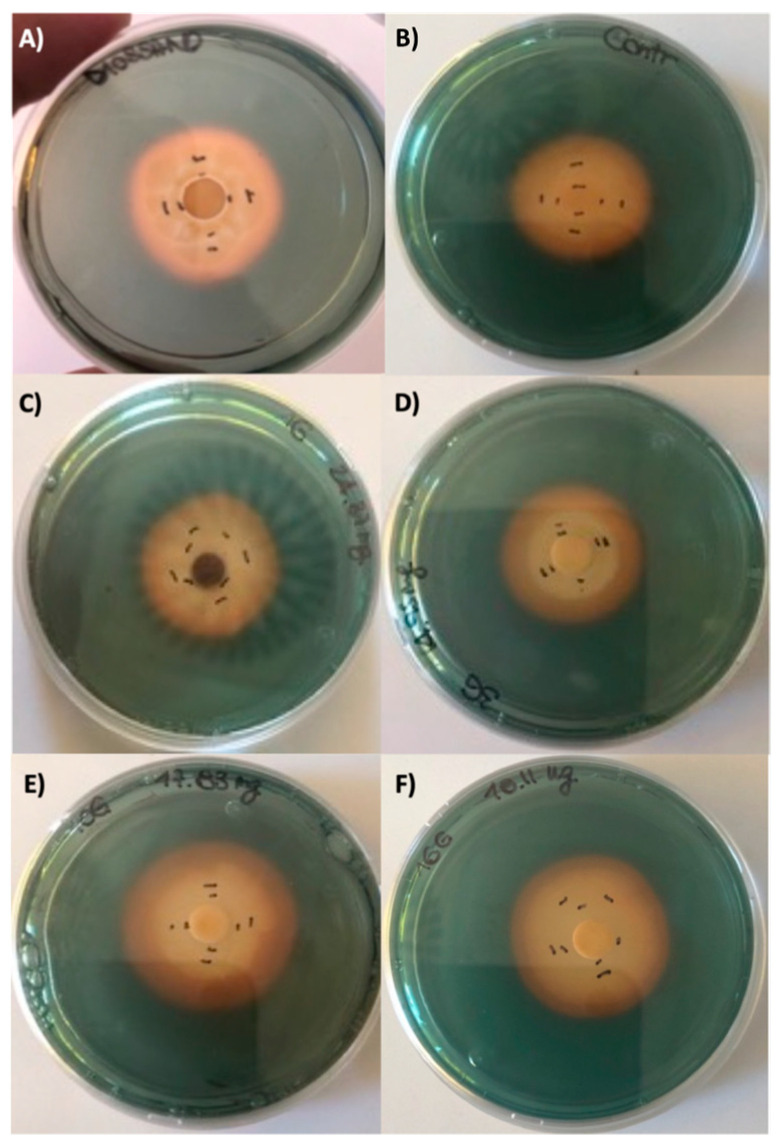
Halo (biological activity) induced by the crude extract obtained from fruiting body of different fungi induced in *P. aeruginosa* ATCC 27853. (**A**) Dioxane (negative control); (**B**) Negative control (only medium); (**C**) *Cortinarius mussivus* (n. 1); (**D**) *C. variecolor* (n. 3); (**E**) *Ramaria pallidosaponaria* (n. 10); (**F**) *Mycena zephyrus* (n. 16). Halo was measured using caliper.

**Figure 7 antibiotics-10-01424-f007:**
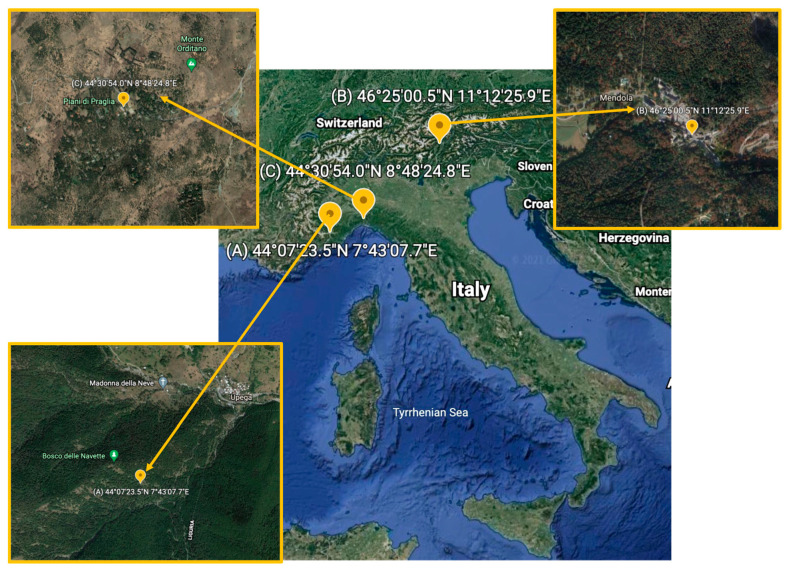
Sampling sites: A (Upega, CN, Italy; 44.1232 N 7.7188 E); B (Passo della Mendola, BZ, Italy; 46.4168 N 11.2072 E); C (Piani di Praglia, GE, Italy; 44.5150 N 8.8069 E).

**Table 1 antibiotics-10-01424-t001:** Fungal species and sampling sites.

Species Number *	Species	Sampling Site ^§^
1	*Cortinarius mussivus*	B
2	*Cortinarius caesiocanescens*	A
3	*Cortinarius variecolor*	B
4	*Cortinarius reverendissimus*	A
5	*Cortinarius percomis*	A
6	*Cortinarius nanceiensis*	B
7	*Cortinarius bovinus*	B
8	*Ramaria pallida*	A
9	*Ramaria parabotrytis*	A
10	*Ramaria pallidosaponaria*	B
11	*Ramaria flavescens*	B
12	*Leucocortinarius bulbiger*	B
13	*Gymnopus confluens*	A
14	*Hydnellum spongiosipes*	C
15	*Mycena renati*	C
16	*Mycena zephyrus*	B

* Species number is a label arbitrarily attributed to each species and it also indicates the crude extract. ^§^ Sampling sites: A (Upega, CN, Italy; 44.1232 N 7.7188 E); B (Passo della Mendola, BZ, Italy; 46.4168 N 11.2072 E); C (Piani di Praglia, GE, Italy; 44.5150 N 8.8069 E).

## Data Availability

The genomic sequences obtained in this study, were available in GenBank with the following entries: SC3_CORT/OL375429, SC1R/OL375430, SC2R /OL375431, SC5/OL375432, SC4_COR/OL375433.

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
