# Peer review of "A Systematic Study of the Antibacterial Activity of Basidiomycota Crude Extracts"

_antibiotics, 2021, doi:10.3390/antibiotics10111424_

Round 1

Reviewer 1 Report

I only find a minor/marginal flaw in the figures needs to be enhanced in future upcoming publications: Conditions of image acquisition needs to be standardized when taking photos about the petri dishes, for the sake of comparability. Some shading and fixed diffuse illumination can eliminate reflections on the photos.

I found the work to be very useful and relevant. Personally, I am looking forward to see the results from the isolated fractions in the next paper. 

Author Response

I only find a minor/marginal flaw in the figures needs to be enhanced in future upcoming publications: Conditions of image acquisition needs to be standardized when taking photos about the petri dishes, for the sake of comparability. Some shading and fixed diffuse illumination can eliminate reflections on the photos.

I found the work to be very useful and relevant. Personally, I am looking forward to see the results from the isolated fractions in the next paper.

The authors thank Reviewer 1 for the precious hints and for the appreciation of their work and they hope to publish the next paper as soon as possible.

Reviewer 2 Report

Dear authors,

The manuscript entitled „A systematic study of the antibacterial activity of Basidiomycota crude extracts” Clericuzio et al. describes the antibacterial and antifungal properties of crude extracts obtained from 16 Basidiomycota against the strain types of five human pathogens. The idea of research was interesting and very good. Conclusions adequate to the conducted research. English language and style are fine.

Several changes are recommended, and some clarifications are required.

First, the authors must carefully read Instructions for authors.

Please clarify figure 4, the letters are too big.

Table 1 in its current form is not suitable.

Author Response

Dear authors,

The manuscript entitled „A systematic study of the antibacterial activity of Basidiomycota crude extracts” Clericuzio et al. describes the antibacterial and antifungal properties of crude extracts obtained from 16 Basidiomycota against the strain types of five human pathogens. The idea of research was interesting and very good. Conclusions adequate to the conducted research. English language and style are fine.

Several changes are recommended, and some clarifications are required.

First, the authors must carefully read Instructions for authors.

Please clarify figure 4, the letters are too big.

The authors modified both figure 4 and figure 5 in order to uniform the font size.

Table 1 in its current form is not suitable.

Table 1 has been modified according to Instructions for authors.

Reviewer 3 Report

This study gives some news inputs regarding the antibacterial and antifungal properties of some Basidiomycota species crude extracts, that can be used against important human pathogens. However, it’s a preliminary study since the assays were only performed with the crude extracts and not with the purified compounds. Nevertheless, the authors refer to this matter in the conclusion.

Introduction

In the introduction some references are referenced with numbers and with names, please correct them.

Line 85: the references should be corrected to [9-13]

Line 92-93: please rewrite the sentence is very confusing.

Line 95-96: the information about a second paper it seems redundant in the introduction.

Results and Discussion

Table 1 has some extra numbers in the left column

Table 1 legend should have more information

Figure 5 should be improved, the axis titles are too small

Figure 6 legend is incorrect , “… C: 1: Cortinarius mussivus; D: 3: C. variecolor; E: 10: Ramaria pallidosaponaria; F: 16: Mycena zephyrus…”

Should be written“…C: Cortinarius mussivus, D: C. variecolor and so on, why the numbers ?

Material and Methods

Line 309:  …”90% acetone – 10% 2-propanol”.   (v/v)?

Line 324: same as line 85

Line 330; correct 6 mm to 6.0 mm

Author Response

This study gives some news inputs regarding the antibacterial and antifungal properties of some Basidiomycota species crude extracts, that can be used against important human pathogens. However, it’s a preliminary study since the assays were only performed with the crude extracts and not with the purified compounds. Nevertheless, the authors refer to this matter in the conclusion.

Introduction

In the introduction some references are referenced with numbers and with names, please correct them.

Done

Line 85: the references should be corrected to [9-13]

Done

Line 92-93: please rewrite the sentence is very confusing.

The authors rewritten this sentence.

Line 95-96: the information about a second paper it seems redundant in the introduction.

The authors agree with the reviewer, deleted this reference to a second paper and modified the conclusions

Results and Discussion

Table 1 has some extra numbers in the left column

The authors modified Table 1 as suggested by reviewer 2

Table 1 legend should have more information

The authors agree with the reviewer, and modified the table.

Figure 5 should be improved, the axis titles are too small

The authors modified Figure 5

Figure 6 legend is incorrect , “… C: 1: Cortinarius mussivus; D: 3: C. variecolor; E: 10: Ramaria pallidosaponaria; F: 16: Mycena zephyrus…”

Should be written“…C: Cortinarius mussivus, D: C. variecolor and so on, why the numbers ?

The numbers referred the label of crude extracts, but, in this form it was not clear, so the authors modified the legend.

Material and Methods

Line 309:  …”90% acetone – 10% 2-propanol”.   (v/v)?

Done

Line 324: same as line 85

Done

Line 330; correct 6 mm to 6.0 mm

Done